# Perturbed and Unperturbed: Analyzing the Conservatively Perturbed Equilibrium (Linear Case)

**DOI:** 10.3390/e22101160

**Published:** 2020-10-15

**Authors:** Yiming Xi, Xinquan Liu, Denis Constales, Gregory S. Yablonsky

**Affiliations:** 1McKelvey School of Engineering, Department of Energy, Environmental and Chemical Engineering, Washington University in St Louis, St. Louis, MO 63130, USA; liuxinquan@wustl.edu (X.L.); gy@wustl.edu (G.S.Y.); 2Department of Electronics and Information Systems, Ghent University, Building S-8, Krijgslaan 281, B-9000 Ghent, Belgium; Denis.Constales@UGent.be

**Keywords:** conservatively perturbed equilibrium, extreme value, momentary equilibrium

## Abstract

The “conservatively perturbed equilibrium” (CPE) technique for a complex chemical system is computationally analyzed in a batch reactor considering different linear mechanisms with three and four species. Contrary to traditional chemical relaxation procedures, in CPE experiments only some initial concentrations are modified; other conditions, including the total amount of chemical elements and temperature are kept unchanged. Generally, for “unperturbed” species with initial concentrations equal to their corresponding equilibrium concentrations, unavoidable extreme values are observed during relaxation to the equilibrium. If the unperturbed species is involved in one step only, this extremum is a momentary equilibrium of the step; if the unperturbed species is involved in more reactions, the extremum is not a momentary equilibrium. The acyclic mechanism with four species may exhibit two extrema and an inflection point, which corresponds to an extremum of the rate of the species change. These facts provide essential information about the detailed mechanism of the complex reaction.

## 1. Introduction

A new kinetic technique, Conservatively Perturbed Equilibrium (CPE), was analyzed theoretically in previous studies [1,2]. It is formulated within a new paradigm of chemical kinetics, the so-called “Joint Kinetics”, which was developed during the last decade (see the papers [3,4,5,6,7,8] and the most recent review [9]). In a batch reactor, in the CPE experiment, the total amounts of chemical elements and the temperature are maintained constant. Then, the CPE procedure is performed as follows:The equilibrium concentration values of all species are determined.Some of the species, at least two, are chosen to have their concentration perturbed from the equilibrium value.At least one species is not chosen, with its concentration value being kept at the equilibrium value.The perturbations mentioned in point 2 are required to satisfy all conservation laws applicable to the system reactions.The evolutions of all species concentrations are observed as they tend towards equilibrium.

It should be stressed that within the CPE procedure, different from the traditional relaxation method, finite perturbations are used, not the small ones. Additionally, the CPE approach is not limited by the linear cases, only by possibilities to meet the balance requirements.

It was shown [1,2] that in such CPE experiments, the concentration of any unperturbed species first evolves away from its initial value, which equals the equilibrium value, and then back to this value as time tends to infinity, via an unavoidable first extreme value (maximum or minimum), possibly followed by other extreme values. These extreme concentration values and the times of their occurrences present essential information:(a)on the detailed mechanism(b)on the values of the kinetic parameters(c)on the possibility of new regimes with an improved yield and selectivity.

The physico-chemical foundations of the CPE technique are the uniqueness and stability of the chemical equilibrium composition, which are basic properties of complex reactions occurring in a closed chemical system. These properties were first qualitatively proven by Zeldovich in 1938 [10,11]; from 1960 onwards, many researchers studied these problems and presented rigorous proofs of the uniqueness and stability of the equilibrium composition, such as Shapiro and Shapley [12], Aris [13,14], Horn and Jackson [15], Vol’pert and Khudyaev [16,17], Gorban [18], and Gorban and Yablonsky [19,20]. An essential trait of the equilibrium of reversible complex reactions is that it is actually a detailed equilibrium—i.e., for every step considered separately, the rate of the forward reaction equals the rate of the corresponding reverse reaction. Reviews of these results are available in the books [21,22] and in the paper [23].

The developed CPE approach is not limited to the networks of only unimolecular, linear reactions which are utilized as examples in the present work. In recent experiments (B. Peng [24]), the CPE technique was experimentally verified in a batch reactor for a nonlinear complex esterification reaction:Alcohol + Acid ⇌ Ester + Ester,
in which ethanol and benzyl alcohol react with acetic acid, producing two different esters and water. Three possible CPE cases (acetic acid and water unperturbed; ethanol and ethyl acetate unperturbed; benzyl alcohol and benzyl acetate unperturbed) were systematically investigated. For the unperturbed species, the unavoidable extreme values were experimentally observed during the relaxation towards equilibrium. These extreme values were larger than the equilibrium concentration. Generally, the CPE technique accompanied by unavoidable extrema may lead to processes limited by thermodynamic equilibrium achieving a yield higher than the equilibrium. This is probably highly important for some industrial processes—for example, processes of the pharmaceutical industry.

In the 1950s and 1960s, fundamental progress in the analysis of chemical relaxations was achieved by Manfred Eigen (Nobel Price 1967 [25]). An excellent review of results of chemical relaxation studies of the 1950–1970s is presented in Bernasconi’s monograph [26]. Typically, in the transient regimes described by Bernasconi, the “perturbed” system goes to the different final composition—i.e., the perturbations of temperature or pressure change the final chemical state. Contrary to classical chemical relaxation procedures, in our CPE approach only some initial concentrations are modified—the “perturbed” ones. Other characteristics, such as the total amounts of chemical elements and the temperature or any other factors that could interfere with the final composition, are kept unchanged to ensure the system returns to the same equilibrium. In the CPE approach, the transient kinetic dependence of unperturbed substances exhibit unavoidable extremum, which are fingerprints of the CPE technique.

### 1.1. Previous Study Review

In the paper [1], the CPE phenomenon was studied computationally and analytically using the two-step linear mechanism as an example
$$A\begin{array}{c} k_{1}{}^{+}\\ ⇄\\ k_{1}{}^{-} \end{array}B\begin{array}{c} k_{2}{}^{+}\\ ⇄\\ k_{2}{}^{-} \end{array}C$$

It exhibits the following properties:

Unavoidability of the concentration extremum of unperturbed species: this fact is independent on the linearity or nonlinearity of the model studied (see Figure 1).

### 1.2. Achieving Momentary Equilibrium (ME) at Some Extrema

Preliminarily, the concept of the “equilibrium” must be explained in more detail with some of its modifications. Typically, the equilibrium is understood as the final state of the non-steady-state chemical reaction which occurs in the closed chemical system.

The “equilibrium of the single step” means that the rate of the forward reaction of this step equals the rate of the corresponding reverse reaction. Consequently, the net rate of this step equals zero, and the equilibrium chemical composition is governed by the equilibrium constant. The “detailed equilibrium” means that every chemical step of the complex chemical reaction at the final state is under equilibrium conditions. As the momentary equilibrium of some reaction, we define such temporal behavior when the rate of some step at that moment equals zero.

At the ME point of a linear step, the quotient of the step (i.e., the ratio of the product concentration to reactant concentration) equals the equilibrium constant. However, the absolute values of the concentrations are not equal to the corresponding concentrations at the final detailed equilibrium.

For systems of linear reactions, this holds for so-called end species, which participate only in a single reaction. For example, in the mechanism
A⇌B⇌C
momentary equilibrium occurs at any extremum of species A and C. In the mechanism
$$\begin{array}{c} A⇌B⇌C\\ ↿⇂\\ D \end{array}$$
momentary equilibrium holds at any extremum of the species A, C and D. For species that participate in several-step reactions the concentration maximum is not a momentary equilibrium.

The time at which the CPE extremum occurs is independent of the perturbation; in the linear mechanism
A⇌B⇌C
these extrema occur at a time that is independent of the magnitude of the perturbation.

The goal of this paper is to study further the dynamic properties of CPE experiments, formulating new questions and new problems, in linear models with three and four species, focusing on the following topics:What is the influence of the mechanism structure on the CPE properties?Which differences exist between noncyclic reactions and cyclic ones? Or four cycles with a diagonal step?What is the influence of the strategy of perturbation—i.e., the distribution of perturbed and unperturbed species, their vicinity and interconnectivity within the mechanism?Is it possible to observe more complex dynamic behavior, such as the evolution of events, two extrema, overshooting the equilibrium value, etc.?

## 2. Materials and Methods

For this study, computational simulations were performed using MATLAB, with ode45 and ode15s as time integration methods for systems of ordinary differential equations that model systems of reactions in batch reactors.

## 3. Results

As mentioned in the introduction, we distinguish two groups of species, “perturbed” and “un-perturbed”. In correspondence with the CPE experiment, the initial concentrations of these species were chosen as equal to the equilibrium concentration (“unperturbed species”) or different from the equilibrium ones (“perturbed” species).

Since the perturbation of a single species cannot be conservative (maintaining the same total amount of each element), and at least one species must remain unperturbed, the number of perturbed species allowed in a CPE experiment ranges from 2 to N-1, where N is the total number of species participating in a mechanism. The number of degrees of freedom of the perturbation equals the number of perturbed species minus the number of independent conservation laws—see [1].

For a linear three-species (N = 3) mechanism, the number of perturbed species is 2, of unperturbed is 3−2 = 1, and there are 2−1 = 1 degrees of freedom. For linear four-species (N = 4) mechanisms, the number of perturbed species can be two or three, but in this study, all perturbations will involve only two species, so that there are 4−2 = 2 unperturbed species and still 2−1 = 1 degrees of freedom.

Additionally, for the purpose of future analysis, it is important to distinguish the subset of special unperturbed species that have only unperturbed neighbors, thus being “shielded” from the perturbation. However, among all species, there must be at least one “neighboring pair” of perturbed and unperturbed species; otherwise, the perturbation would have no effect on the unperturbed species.

### 3.1. Analysis of Perturbed Species in a Three-Species Acyclic Mechanism (Two-Step Mechanism)

$$A\begin{array}{c} k_{1}{}^{+}\\ ⇄\\ k_{1}{}^{-} \end{array}B\begin{array}{c} k_{2}{}^{+}\\ ⇄\\ k_{2}{}^{-} \end{array}C$$

The previous work [1] discussed the transient behavior of the unperturbed species only in two-step mechanisms. As a continuation of that analysis, we now consider the perturbed species. A series of simulation experiments with different kinetic constants was performed, from which two experiments were selected as examples—see Table 1.

The transient trajectories of the three species in the two experiments are plotted in Figure 2.

#### *New Findings*—Perturbed Species May Experience Either Monotone Relaxation or Behavior with one Extremum Peak

Changing the kinetic constants may modify the extremum behavior for the perturbed species. In this case, species A and B both exhibit monotone relaxation when k_1_^+^ is chosen in a small range. As the forward kinetic constant becomes larger, species B shows transition to more complex behavior. Its transient regime will come to contain one peak (extremum). Additionally, the concentration of B will overshoot its equilibrium value (in the graphs, this is where conversion = 0).

The concentration extremum indicates the rate of change of the corresponding species when the timepoint is zero. Since this behavior occurs on species B, which is participating in two reactions simultaneously, this extremum is not a momentary equilibrium. It was discovered in a series of computational simulations for this mechanism that when the kinetic constant is above some value, a concentration extremum for species B will become available.

### 3.2. Three-Species Cyclic Mechanism

The three-species cyclic mechanism (see Figure 3 and Table 2) presents an additional connectivity from species A to species C. The kinetic constants must satisfy the following Onsager condition: the ratio of the third pair of kinetic constants, $\frac{k_{3}^{+}}{k_{3}^{-}}$, which is also the thermodynamic equilibrium constant of the third reaction K_3_, must satisfy the condition $\frac{k_{1}^{+}}{k_{1}^{-}}\cdot \frac{k_{2}^{+}}{k_{2}^{-}}=\frac{k_{3}^{+}}{k_{3}^{-}}$.

Experiments were performed as described in Table 2 and plotted in Figure 4, setting the same first two pairs of reaction parameters as in Experiment 2 above, and in adding the third reaction with kinetic constants satisfying the Onsager condition.

#### 3.2.1. *New Findings on the Three-Species Cyclic Mechanism*—The Extremum Time for the Cyclic Mechanism Has the Same Analytical Expression as for the Acyclic

Since a three-species mechanism has three eigenvalues, one of which is 0, and the discrepancy from equilibrium of an unperturbed substance is zero both at times zero and infinity, it must be proportional to the difference $e^{\lambda_{p}t}-e^{\lambda_{m}t}$. Therefore, the extremum time, when its derivative with respect to time vanishes, is given by
$$t_{exe}=\frac{log\left(\lambda_{p}/\lambda_{m}\right)}{\lambda_{m}-\lambda_{p}},$$
which is the same as those in the acyclic mechanism case. This is because there are only two nonzero eigenvalues in any linear three-species system, and since there are two boundary conditions at *t* = 0 and as *t* approaches infinity, the linear combinations of exponentials are fixed up to a factor for the unperturbed species.

#### 3.2.2. The Cyclic Mechanism’s Extremum Time is Shorter than that of the Acyclic Mechanism

The cyclic and acyclic exhibit similar properties, however, the major difference between them is the extremum time caused by the reaction between A and C. As mentioned before, the cycle must satisfy the Onsager relation $\frac{k_{1}^{+}}{k_{1}^{-}}\cdot \frac{k_{2}^{+}}{k_{2}^{-}}=\frac{k_{3}^{+}}{k_{3}^{-}}$, otherwise, there are possibilities of complex eigenvalues. To compare the cyclic and the acyclic mechanisms, a simulation is performed fixing k_1_^+^ = 0.16 s^−1^, k_1_^−^ = 0.04 s^−1^, k_2_^+^ = 0.12 s^−1^, k_2_^−^ = 0.06 s^−1^, and selecting the value of k_3_^+^, which in turn determines the value of k_3_^−^ from the thermodynamic condition. The comparison results are illustrated in Figure 5.

The plot on the left is when we have a large k_3_^+^ and the plot on the right is when have a small k_3_^+^. We observe that with a larger k_3_^+^, the extremum time becomes smaller and the extremum concentration gets closer to the equilibrium concentration. With a smaller k_3_^+^, the trajectories of a cyclic mechanism become closer to that of an acyclic mechanism with the same k_1_^+^, k_1_^−^, k_2_^+^, and k_2_^−^ values. Intuitively, we can consider the reaction between species A and species C as a shortcut, so that a small k_3_^+^ will keep the reaction going similar to an acyclic, but a large k_3_^+^ will allow the reaction to reach equilibrium faster through this shortcut. Mathematically, the reaction between species A and species C changes the zero entries on the upper right and lower left corner of the kinetic matrix to k_3_^+^ and k_3_^−^, as can be seen from the matrix form of the kinetic model:$$\frac{d}{dt}\left(\begin{array}{c} \left[A\right]\\ \left[B\right]\\ \left[C\right] \end{array}\right)=\left(\begin{array}{ccc} -k_{1}^{+}-k_{3}^{-} & k_{1}^{-} & k_{3}^{+}\\ k_{1}^{+} & -k_{1}^{-}-k_{2}^{+} & k_{2}^{-}\\ k_{3}^{-} & k_{2}^{+} & -k_{3}^{+}-k_{2}^{-} \end{array}\right)\left(\begin{array}{c} \left[A\right]\\ \left[B\right]\\ \left[C\right] \end{array}\right)$$

A detailed proof of the faster evolution in a three-species cyclic mechanism is given in Appendix A.

There is No momentary equilibrium due to no single-step species.

All three species are involved in two steps, which means that they cannot exhibit a momentary equilibrium in any of the steps. Nevertheless, for this three-species cyclic mechanism the concentration extremum will occur for unperturbed species during the transient regimes, and some special cases such as unitary kinetic parameters would generate special results. This feature will be explored in a future paper with focus on CPE for reaction systems with special parameters.

### 3.3. Four-Species Acyclic Mechanism

The results of this case are summarized in Figure 6 and Figure 7 and Table 3.

#### 3.3.1. *New Findings*—Possibility of Two Extrema and an Inflection Point

Unperturbed species can exhibit two concentration extrema during CPE, a maximum and a minimum, located before and after the point of crossing the equilibrium value in the transient regime. Which extremum occurs first depends on the sign of the perturbations. The times at which the extrema are reached are not dependent upon initial conditions. An inflection point occurs between them, where the second derivative becomes zero for the unperturbed species concentration; the physical meaning is a maximum or minimum of the rate of the substance.

This mechanism leads to six different cases of perturbation when choosing two unperturbed species among four. The simulation settings for all the six combinations are listed in Table 4 and Table 5, and two representative cases are plotted in Figure 8. In these experiments, all simulations are performed under the same parameters; only the choice of perturbed species varies.

#### 3.3.2. The Initial Rate is Zero for Unperturbed Species that are Connected only with Other Unperturbed Species

In the cases where species A and B are perturbed (i.e., C, D unperturbed), the initial slope of the trajectory of species D is 0 (rate of reaction is 0). The same phenomenon is also observed in species A when C and D are perturbed (A and B unperturbed). See Figure 9.

In these cases, when the unperturbed species is not connected directly with perturbed species, the initial momentary equilibrium established between the unperturbed species produces a zero initial rate for the species, making the species appear “shielded” by the other species. Nevertheless, this momentary equilibrium will immediately be broken as the other species participate in reactions with perturbed species. The first derivatives of concentrations (i.e., reaction rates) in experiment #1 are plotted in Figure 10. When an initial rate is zero, the extremum rate is unavoidable since the final rate is zero as well. Consequently, an inflexion point of transient concentration is unavoidable.

#### 3.3.3. Evolution of Events: Change of the Number of Extrema due to Change in Kinetic Parameter Values

In Experiment #3, species C has two extrema and one inflection point under the given kinetic parameter set. Varying a kinetic parameter such as k_1_^+^ will lead to a change in the number of extrema, as is plotted in Figure 11.

This shows that a threshold located between k_1_^+^ = 1.5 and k_1_^+^ = 1.6 acts as bifurcation point where switching from the one-extremum to the two-extrema case. This passage from one case to the other may be helpful to identify the source of complexity in this linear dynamic system. Similar bifurcations are discovered for parameters k_1_^−^, k_2_^+^ and k_3_^−^. These questions will motivate future studies of the evolution of events in CPE experiments.

### 3.4. Four-Species Cyclic Mechanism

The mechanism is shown in Figure 12. The parameter values used for the simulations are listed in Table 6, along with the choices of perturbed species. Example time evolutions are plotted in Figure 13.

#### 3.4.1. *New Findings*—Similarity with Four-Species Acyclic Mechanism: Occurrence of Two Extrema and an Inflection Point

Since the four-species acyclic mechanism is a limit of four-species cyclic mechanisms, it is not surprising to find that the four-species cyclic CPE shares similarity in transient behavior for unperturbed species. Unperturbed species can also exhibit two extrema and one inflection point, as well as crossing their equilibrium concentration during the transient regime. The additional connectivity did not overly modify the complexity of the system behavior.

#### 3.4.2. Zero Initial Rate Behavior for Unperturbed Species Does not Occur

The four-species cyclic mechanism no longer has “end” species participating in only one step; the example plots of rates in Figure 14 illustrates a case where species C and D are unperturbed.

### 3.5. Four-Species Cyclic Mechanism with Additional Diagonal Connectivity

To investigate further the effect of the mechanism on the behavior of CPE trajectories, we consider systems that are square but connected diagonally, such as the mechanism shown in Figure 15. It is made of two triangles joined into a square, which means that it must satisfy three Onsager relationships: $\frac{k_{1}^{+}}{k_{1}^{-}}\cdot \frac{k_{2}^{+}}{k_{2}^{-}}\cdot \frac{k_{3}^{+}}{k_{3}^{-}}=\frac{k_{4}^{+}}{k_{4}^{-}},\;\frac{k_{1}^{+}}{k_{1}^{-}}\cdot \frac{k_{2}^{+}}{k_{2}^{-}}=\frac{k_{5}^{+}}{k_{5}^{-}},\;\mathrm{and}\;\frac{k_{5}^{+}}{k_{5}^{-}}\cdot \frac{k_{3}^{+}}{k_{3}^{-}}=\frac{k_{4}^{+}}{k_{4}^{-}}$, of which only two are independent.

An interesting question could be asked regarding the effect of $k_{5}^{+}$ (and $k_{5}^{-}$) on the trajectories. In Figure 16, we see that a system with a larger $k_{5}^{+}\;$ will behave more akin to a triangular mechanism (or two triangular mechanisms) while a system with a smaller $k_{5}^{+}$ will behave more akin to a square. In particular, when $k_{5}^{+}$ is large, the two extrema become only one.

## 4. Discussion

### 4.1. Comparing Structural Differences: Number of Species in Mechanism

Four-species mechanisms, in comparison with three-species mechanisms, demonstrate additional complex behavior (i.e., two extrema in concentration instead of one, and an inflexion point in concentration, which is an extremum in rate). Although the mechanisms in the current studies are all linear systems, a series of observable transient complexities can still be interesting to show further detailed information on the mechanism.

### 4.2. Evolution of Events: Effects of Kinetic Parameters on Complexity

Some complex behaviors only occur in parameter subdomains. Finding the boundary point between a lower and higher complexity (e.g., transition from one extremum to two) remains challenging because it entails the analysis of linear combinations of exponentials, which becomes complicated beyond three-species mechanisms. Overall, the importance of effects determined by kinetic parameters will be an important aspect in further studies of CPE experiments.

## 5. Conclusions and Future Applications of CPE

The conservatively perturbed equilibrium technique was studied for a given set of chemical mechanisms, featuring acyclic and cyclic mechanisms involving three or four species. Additional structures with different connectivities were also analyzed. The CPE approach allowed to find a link between the type of mechanism and properties of the system relaxation, contributing to decoding the behavior of dynamical systems and relating the observed phenomena to the underlying complexities.

When an unperturbed species is not connected directly with perturbed species, the initial rate is zero, and a rate extremum is unavoidable, which is an inflexion point of transient concentration.

Future applications of CPE could be in catalysis, in simulating isotope exchange reactions, or even more widely, in analyzing large biological systems.

## Figures and Tables

**Figure 1 entropy-22-01160-f001:**
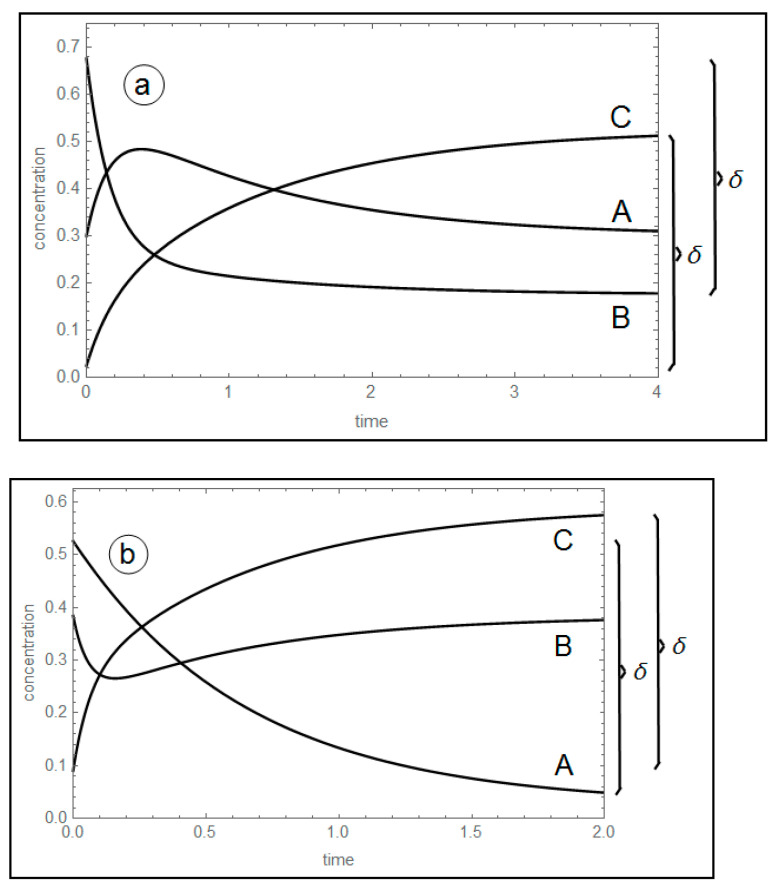
Concentration profiles of A, B and C, for δ = 0.5. (**a**) A maximum of A; k_1_^+^ = 1.75, k_1_^−^ = 3.00, k_2_^+^ = 1.50, k_2_^−^ = 0.50 s^−1^. (**b**) A minimum of B; k_1_^+^ = 1.50, k_1_^−^ = 0.10, k_2_^+^ = 10.0, k_2_^−^ = 6.50 s^−1^.

**Figure 2 entropy-22-01160-f002:**
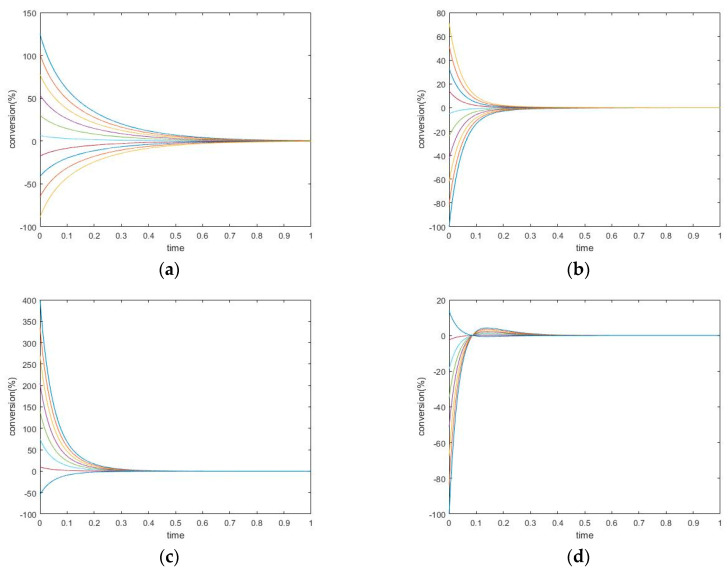
Two-step mechanism, (**a**,**b**) simulation experiment #1, (**c**,**d**) simulation experiment #2. (**a**) Species A Conversion, (**b**) Species B Conversion; (**c**) Species A Conversion; (**d**) Species B Conversion.

**Figure 3 entropy-22-01160-f003:**
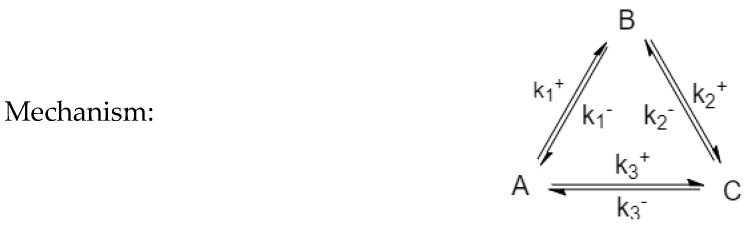
Triangular (three-species cyclic) mechanism.

**Figure 4 entropy-22-01160-f004:**
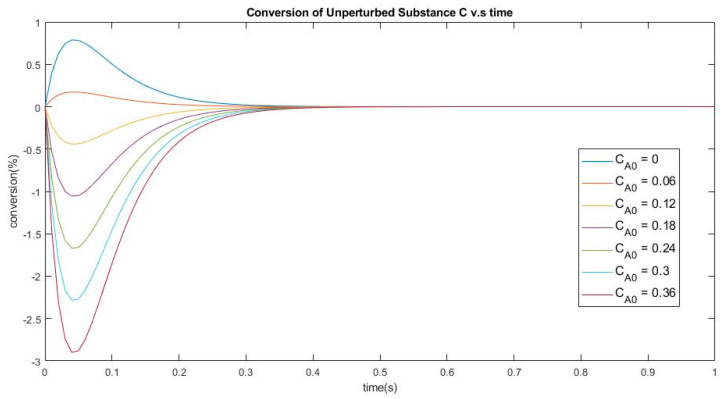
Conversion of the unperturbed species C vs. time. The perturbation consists in changing species (A) from 0 to 0.36 and species (B) from 0.92 down to 0.56.

**Figure 5 entropy-22-01160-f005:**
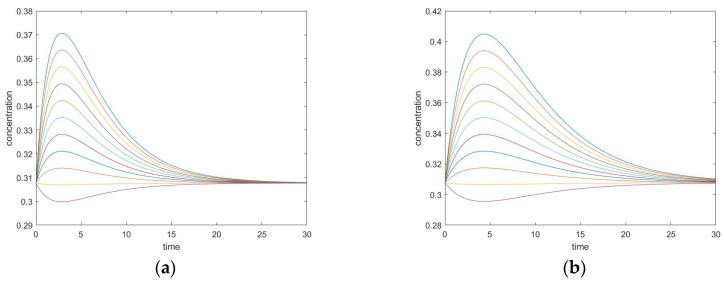
(**a**) Conversion of unperturbed species with large k_3_^+^. (**b**) Conversion of unperturbed species with small k_3_^+^.

**Figure 6 entropy-22-01160-f006:**
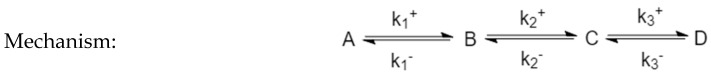
Four-species acyclic mechanism.

**Figure 7 entropy-22-01160-f007:**
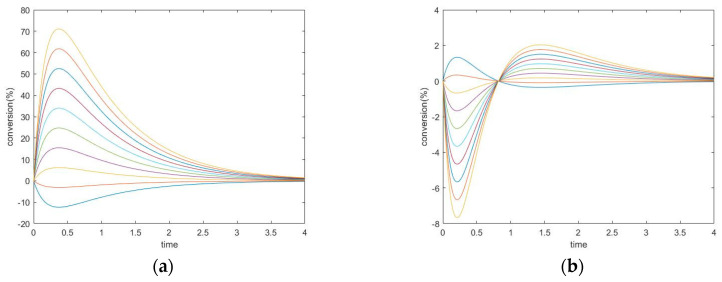
Four-species acyclic mechanism, species B and C unperturbed, (**a**) conversion of unperturbed species B, (**b**) conversion of unperturbed species C.

**Figure 8 entropy-22-01160-f008:**
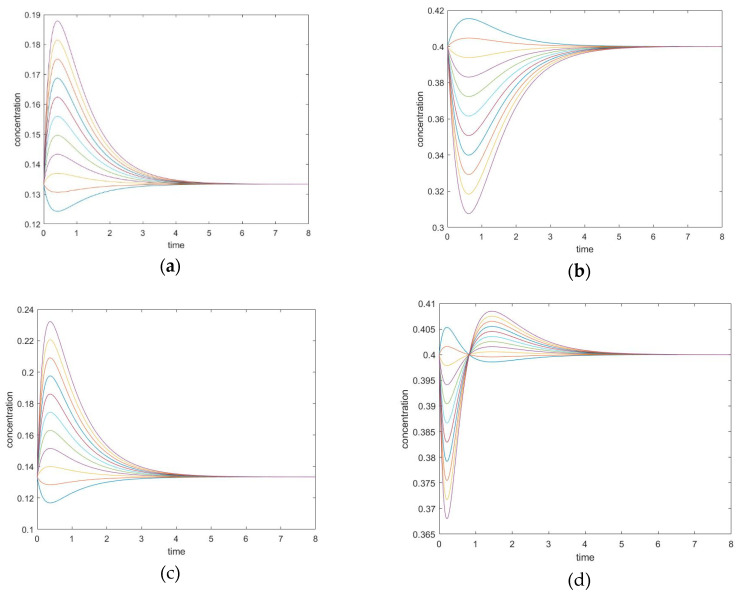
(**a**,**b**) Experiment 2, (**a**) Concentration of the unperturbed species B, and (**b**) concentration of the unperturbed species D. (**c**,**d**) Experiment 3, (**c**) Concentration of the unperturbed species B, (**d**) concentration of the unperturbed species C.

**Figure 9 entropy-22-01160-f009:**
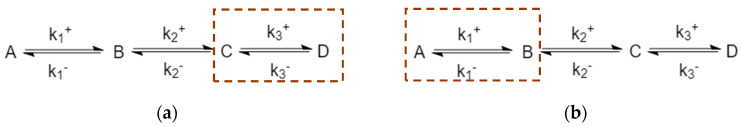
(**a**) Species C and D are unperturbed in experiment #1. (**b**) Species A and B are unperturbed in experiment #6.

**Figure 10 entropy-22-01160-f010:**
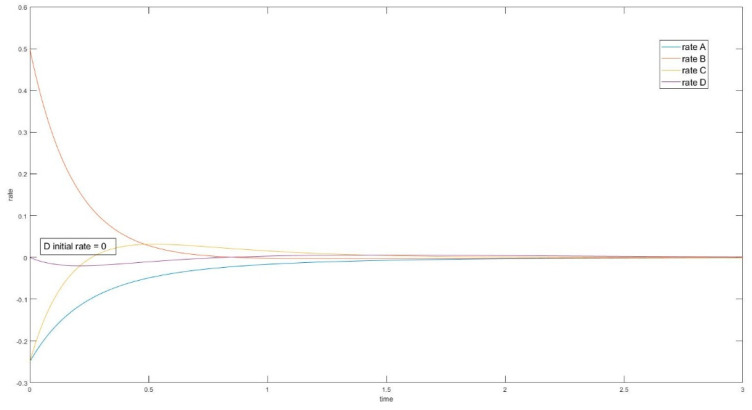
Four-species acyclic reaction, species C and D unperturbed, first derivative. Species D has initial rate equal to 0.

**Figure 11 entropy-22-01160-f011:**
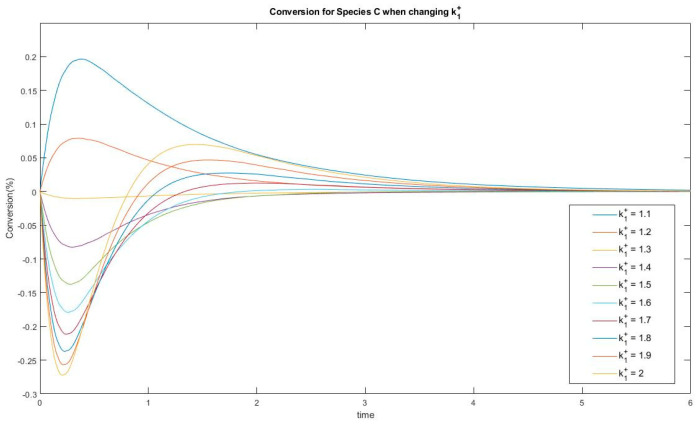
Conversion of species C when changing k_1_^+^. A passage from one extremum to two extrema behavior is observed.

**Figure 12 entropy-22-01160-f012:**
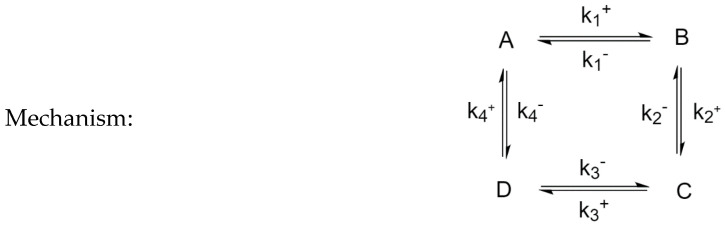
Four-Species Cyclic Mechanism.

**Figure 13 entropy-22-01160-f013:**
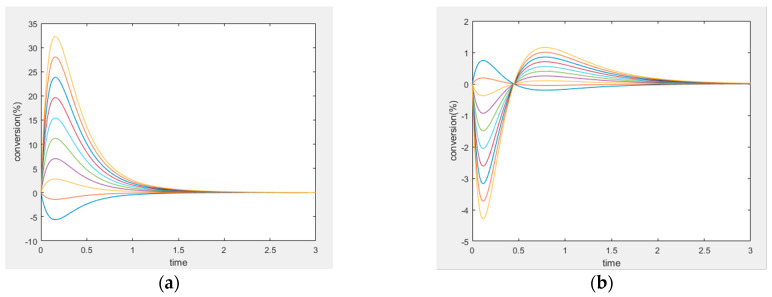
Four-species cyclic mechanism, BC unperturbed, (**a**) conversion of unperturbed species B, (**b**) conversion of unperturbed species C.

**Figure 14 entropy-22-01160-f014:**
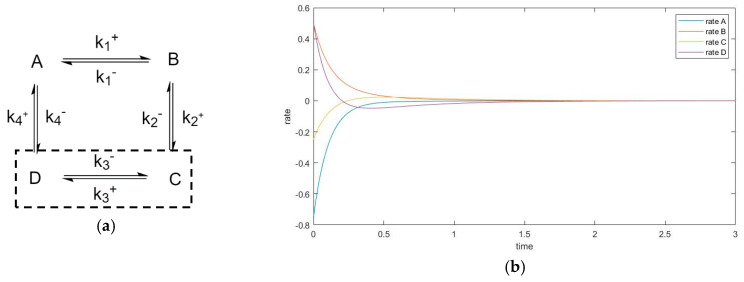
(**a**) Species C and D are unperturbed in the four-species cyclic mechanism experiment. (**b**) plot of first derivatives (rates). Zero initial rate does not occur.

**Figure 15 entropy-22-01160-f015:**
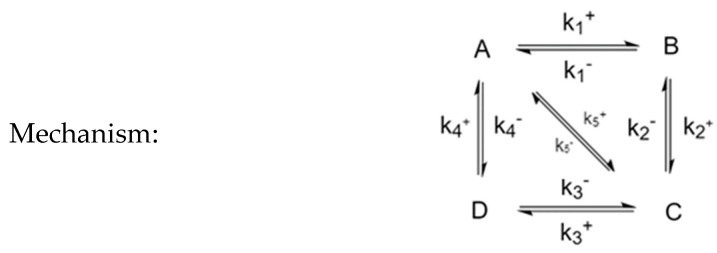
Four-species cyclic mechanism with diagonal connectivity.

**Figure 16 entropy-22-01160-f016:**
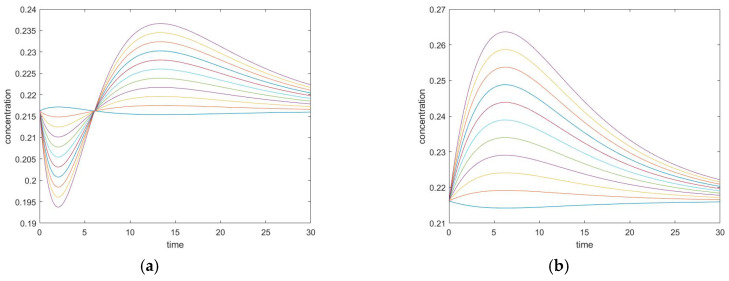
(**a**) Small diagonal kinetic constant $k_{5}^{+}$, (**b**) large diagonal kinetic constant $k_{5}^{+}$.

**Table 1 entropy-22-01160-t001:** Two-step mechanism, Simulation Experiments 1 and 2.

Experiment Settings	Experiment #1	Experiment #2
Kinetic Parameters (s^−1^):	k_1_^+^ = 5, k_1_^−^ = 4 k_2_^+^ = 12, k_2_^−^ = 6	k_1_^+^ = 16, k_1_^−^ = 4 k_2_^+^ = 12, k_2_^−^ = 6
Perturbed species:	A, B	A, B
Unperturbed species:	C	C

**Table 2 entropy-22-01160-t002:** Three-species cyclic conservatively perturbed equilibrium (CPE) example.

Experiment Settings	Value	
Kinetic Parameters (s^−1^):	k_1_^+^ = 16	k_1_^−^ = 4
	k_2_^+^ = 12	k_2_^−^ = 6
	k_3_^+^ = 8	k_3_^−^ = 1
Perturbed species:	A, B	
Unperturbed species:	C	

**Table 3 entropy-22-01160-t003:** A four-species acyclic example.

Experimental Settings	Value	
Kinetic parameters (s^−1^):	k_1_^+^ = 2	k_1_^−^ = 1
	k_2_^+^ = 3	k_2_^−^ = 1
	k_3_^+^ = 1	k_3_^−^ = 1
Perturbed species:	A, D	
Unperturbed species:	B, C	

**Table 4 entropy-22-01160-t004:** Experiment settings of CPE for a four-species acyclic mechanism.

Experimental Settings	Values	
Kinetic parameters (s^−1^):	k_1_^+^ = 2	k_1_^−^ = 1
	k_2_^+^ = 3	k_2_^−^ = 1
	k_3_^+^ = 1	k_3_^−^ = 1

**Table 5 entropy-22-01160-t005:** Cases and results of CPE for a four-species acyclic mechanism.

Experiment	Perturbed Species	Unperturbed Species	Behavior
1	A, B	C, D	2 extrema of [C], 1 of [D]
2	A, C	B, D	1 extremum of [B], 1 of [D]
3	A, D	B, C	2 extrema of [C], 1 of [B]
4	B, C	A, D	1 extremum of [A], 1 of [D]
5	B, D	A, C	1 extremum of [A], 1 of [C]
6	C, D	A, B	1 extremum of [A], 1 of [B]

**Table 6 entropy-22-01160-t006:** Kinetic specs of four-species cyclic mechanism.

Experiment Settings	Value	
Kinetic parameters (s^−1^):	k_1_^+^ = 2	k_1_^−^ = 1
	k_2_^+^ = 3	k_2_^−^ = 1
	k_3_^+^ = 1	k_3_^−^ = 1
	k_4_^+^ = 1	k_4_^−^ = 6
Perturbed species:	A, D	
Unperturbed species:	B, C	

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
