# Peer review of "Perturbed and Unperturbed: Analyzing the Conservatively Perturbed Equilibrium (Linear Case)"

_entropy, 2020, doi:10.3390/e22101160_

Round 1

Reviewer 1 Report

General remarks:

The paper describes an interesting idea for getting information on rate constants for
chemical systems from measuring the concentration response from concentration
perturbations of the equilibrium state.

One problem for the general applicability is, that the method as presented in this paper
is restricted to networks of unimolecular reactions, which are a minor subset of real
chemical networks.

A more serious problem is, that the investigated perturbations experimentally involve
an increase and decrease of the concentration of a particular species without changing
anything else. An increase of a concentration is easily achieved, but the paper should
explain how a decrease of a concentration is done in an actual experiment?

Specific remarks:

l63: reference 24 describing an actual experiment could not be found.

l65: give references for the three experiments, and show some results.

l69: More detailed explanation required.

l104: ... momentary detailed balance of that single reaction.

l200: The perturbation consists in changing [A] from 0 to 0.36 and [B] correspondingly.
(In chemistry A means species A and [A] means concentration of A.

l208: Explain how this his expression is derived.

l224: Which relaxation time.

l225: ... gets closer to the equilibrium concentration.

l230: What is the kinetic matrix?

l265: Behavior column: 1 extremum on what?

Reviewer 2 Report

This is a useful paper, but its scope is relatively narrow. I believe the mathematical details and the kinetic conclusions are entirely correct. However, the authors do not mention the fact that experimental chemical kinetics has already used conservatively perturbed equilibrium experiments for more than 50 years now: the techniques temperature jump, pressure jump and filed jump all use these principles. There are a lot of papers focusing on the mathematical details of these relaxation experiments, even a widely used book was written on them in the mid 1970s (C. Bernasconi: Relaxation Kinetics, Academic Press, New York 1976.). I very much recommend to the authors that they overview this earlier knowledge and present their new work in its context.

Minor Points:
Point 5 is empty in line 129 (page 4).
Year, volume, and page numbers are missing from reference 3.
The states of references 9 and 24 should be updated if possible
